# Disruption of O-GlcNAcylation Homeostasis Induced Ovarian Granulosa Cell Injury in Bovine

**DOI:** 10.3390/ijms23147815

**Published:** 2022-07-15

**Authors:** Teng-Fei Wang, Zhi-Qiang Feng, Ya-Wen Sun, Shan-Jiang Zhao, Hui-Ying Zou, Hai-Sheng Hao, Wei-Hua Du, Xue-Ming Zhao, Hua-Bin Zhu, Yun-Wei Pang

**Affiliations:** Embryo Biotechnology and Reproduction Laboratory, Institute of Animal Science, Chinese Academy of Agricultural Sciences, Beijing 100193, China; wangtengfei0714@163.com (T.-F.W.); zhiqiangg_feng@163.com (Z.-Q.F.); sunyw0917@163.com (Y.-W.S.); zhaoshanjiang@caas.cn (S.-J.Z.); zouhuiying@caas.cn (H.-Y.Z.); haohaisheng@caas.cn (H.-S.H.); duweihua@caas.cn (W.-H.D.); zhaoxueming@caas.cn (X.-M.Z.); zhuhuabin@caas.cn (H.-B.Z.)

**Keywords:** O-GlcNAcylation, OGT, OGA, granulosa cells, glucose, TXNIP, bovine

## Abstract

O-linked β-N-acetylglucosamine (O-GlcNAc) modification is a ubiquitous, reversible, and highly dynamic post-translational modification, which takes charge of almost all biological processes examined. However, little information is available regarding the molecular regulation of O-GlcNAcylation in granulosa cell function and glucose metabolism. This study focused on the impact of disrupted O-GlcNAc cycling on the proliferation and apoptosis of bovine granulosa cells, and further aimed to determine how this influenced glucose metabolism. Pharmacological inhibition of OGT with benzyl-2-acetamido-2-deoxy-α-D-galactopyranoside (BADGP) led to decreased cellular O-GlcNAc levels, as well as OGT and OGA protein expressions, whereas increasing O-GlcNAc levels with the OGA inhibitor, O-(2-acetamido-2-deoxy-D-gluco-pyranosylidene) (PUGNAc), resulted in elevated OGA protein expression and decreased OGT protein expression in granulosa cells. Dysregulated O-GlcNAc cycling reduced cell viability, downregulated the proliferation-related genes of *CDC42* and *PCNA* transcripts, upregulated the pro-apoptotic genes of *BAX* and *CASPASE-3* mRNA and the ratio of *BAX/BCL-2*, and increased the apoptotic rate. Glycolytic enzyme activities of hexokinase and pyruvate kinase, metabolite contents of pyruvate and lactate, mitochondrial membrane potential, ATP levels, and intermediate metabolic enzyme activities of succinate dehydrogenase and malate dehydrogenase involved in the tricarboxylic acid cycle, were significantly impaired in response to altered O-GlcNAc levels. Moreover, inhibition of OGT significantly increased the expression level of thioredoxin-interacting protein (TXNIP), but repression of OGA had no effect. Collectively, our results suggest that perturbation of O-GlcNAc cycling has a profound effect on granulosa cell function and glucose metabolism.

## 1. Introduction

Cumulus and mural granulosa cells (GCs) of the ovarian follicle surround and interact with the developing oocyte. These follicular cells reflect the characteristics of the oocyte and play a vital role in regulating oocyte maturation [1]. Generally, proliferation and differentiation of GCs leads to follicular maturation and ovulation, whereas apoptosis within GCs results in poor ovarian responsiveness to gonadotrophin stimulation and, consequently, the degeneration of oocytes [2]. Therefore, a proper regulation of GC proliferation is of ultimate importance for follicle growth and the provision of a unique microenvironment for oocyte maturation [3]. In fact, GCs provide the oocytes with essential nutrients to meet their requirements [4].

O-linked β-N-acetylglucosamine (O-GlcNAc) modification (O-GlcNAcylation) is a ubiquitous, reversible, and highly dynamic post-translational modification that was found on serine and threonine (Ser/Thr) residues of intracellular nuclear, cytoplasmic, and mitochondrial proteins [5,6,7]. At the cellular level, O-GlcNAcylation takes charge of almost all biological processes examined [8,9]. Thus far, over 7000 known protein sites are modified by O-GlcNAcylation, with more proteins and sites being routinely identified with the development in detection [10,11,12]. O-GlcNAcylation is dependent on the nutrient flux through the hexosamine biosynthetic pathway (HBP), which reflects the availability of nutrients and energy [13,14]. Cellular and environmental stressors, such as osmotic, reactive oxygen species, heat shock, hypoxia, or toxins, could all affect the cellular nutrient supply, thus resulting in aberrant O-GlcNAcylation [12]. Unlike other glycosylation, O-GlcNAcylation is regulated only by two enzymes: O-GlcNAc transferase (OGT), which covalently adds an O-GlcNAcylation moiety to Ser/Thr residues of target proteins, and O-GlcNAcase (OGA), which reverses this process [15]. This single pair of enzymes is coordinately regulated to maintain the overall cellular and tissue O-GlcNAcylation balance in respond to environmental changes [16].

Emerging evidence suggests that O-GlcNAc cycling has profound effects on the reproductive system [17]. In mouse deletion models, researchers found that OGT deletion results in the loss of embryonic stem cell viability [18], and genetic disruption of OGA also leads to developmental defects during embryogenesis [19]. O-GlcNAcylation occurs in oocytes during meiosis [20], and its perturbation significantly compromises zygote development [21]. Several implantation processes, such as trophoblast proliferation, trophectoderm differentiation, and embryonic cell differentiation, are affected by O-GlcNAcylation as well [7,22].

Glucose is an important metabolic substrate and preferably utilized by GCs to provide the oocytes with energy production [23]. A recent study demonstrated that O-GlcNAcylation varies in different sizes of antral follicles and influences GC proliferation in bovines [24]. However, the detailed molecular regulation of O-GlcNAcylation in GC function is just beginning to be clarified. This study was therefore conducted to explore the impact of disrupted O-GlcNAc cycling on the proliferation and apoptosis of bovine GCs, and further aimed to determine how this influenced glucose metabolism.

## 2. Results

### 2.1. Altered O-GlcNAc Levels Affect the Expression of OGT and OGA in Bovine GCs

To investigate the feedback regulations for the maintenance of cellular O-GlcNAc homeostasis in bovine GCs, an OGT inhibitor, benzyl-2-acetamido-2-deoxy-α-D-galactopyranoside (BADGP) (Sigma-Aldrich, St. Louis, MO, USA), or an OGA inhibitor, O-(2-acetamido-2-deoxy-D-gluco-pyranosylidene) (PUGNAc) (Sigma-Aldrich, St. Louis, MO, USA) was used to alter cellular O-GlcNAc levels in this study. The first generation of GCs were randomly divided into three groups, and treated for 24 h without or with BADGP (4 mm) or PUGNAc (100 μM). The concentrations of BADGP and PUGNAc were chosen based on our preliminary studies (Appendix A) and a previously published study, respectively [25]. The O-GlcNAc levels were significantly decreased in the BADGP-treated samples, whereas a robust increase in the global O-GlcNAcylation content was observed in GCs exposed to PUGNAc compared with the control group (*p* < 0.05; Figure 1A). When GCs were treated with BADGP or PUGNAc, OGT mRNA and protein levels were markedly downregulated compared to the control (*p* < 0.05; Figure 1B,D). However, the *OGA* transcript level was dramatically elevated after BADGP or PUGNAc treatment compared to control cells without any treatment (*p* < 0.05; Figure 1C). The OGA protein level corresponded with the increase in mRNA levels following PUGNAc exposure, but BADGP-treated GCs showed a decrease in the OGA protein level compared to the control (*p* < 0.05; Figure 1E).

### 2.2. Disruption of O-GlcNAc Cycling Affects Viability and Proliferation of GCs

The proliferative potential of GCs was further evaluated in order to verify whether the altered O-GlcNAc level is involved in this process. As illustrated in Figure 2, a significant reduction in cell viability was observed in samples challenged with BADGP or PUGNAc at 12 h and 24 h compared to the untreated control (*p* < 0.05; Figure 2A,B). Next, the relative expression levels of proliferation-associated genes (*PCNA*, *CDC42*, and *CCND2*) were measured. Results showed that the effective disruption of O-GlcNAc cycling following BADGP or PUGNAc exposure resulted in lower mRNA transcripts of *PCNA* and *CDC42* at different degrees, whereas the expression of *CCND2* was unaffected by the treatments (Figure 2C).

### 2.3. Disruption of O-GlcNAc Cycling Induces Cell Apoptosis in Bovine GCs

Given that dysregulated O-GlcNAcylation is often involved in various cellular functions, including apoptosis, we examined whether changes in O-GlcNAc levels influenced apoptosis in bovine GCs. Flow cytometric analysis indicated that the BADGP or PUGNAc treatment significantly increased the number of apoptotic cells (*p* < 0.05; Figure 3A,B). To further confirm our results, the mRNA transcript levels of genes associated with apoptosis were measured. As shown in Figure 3, marked inductions of *BAX* and *CASPASE-3* at the mRNAs were observed upon BADGP or PUGNAc exposure. The *BCL-2* transcript was comparable with that in the control group (*p* > 0.05), but altered O-GlcNAc levels resulted in a significant increase in the apoptotic-related *BAX/BCL-2* mRNA ratio (*p* < 0.05; Figure 3C).

### 2.4. Perturbation of O-GlcNAc Cycling in Bovine GCs Reduces Glycolysis

To analyze whether BADGP- or PUGNAc-induced perturbation of O-GlcNAc cycling impacts glucose metabolism in bovine GCs, we measured several essential components of the glycolysis, including activities of hexokinase (HK), pyruvate kinase (PK), and lactate dehydrogenase (LDH), as well as pyruvate and lactate production. As illustrated in Figure 4, the activity of HK in PUGNAc-exposed samples was much lower than that in the control and BADGP treatment group (*p* < 0.05; Figure 4A). Upon BADGP or PUGNAc exposure, GCs exhibited reduced PK activity (Figure 4B) and pyruvate (Figure 4D) and lactate production (Figure 4E) relative to the control samples, whereas the enzymatic activity of LDH in GCs was significantly increased (*p* < 0.05; Figure 4C).

### 2.5. Alteration in O-GlcNAc Levels Impairs Mitochondria Homeostasis, ATP Production, and the Tricarboxylic Acid (TCA) Cycle

To find out whether altered O-GlcNAc levels change the mitochondrial membrane potential (MMP) and ATP production, bovine GCs were first stained with JC-1, a reporter molecule for heterogeneity in MMP in living cells. Flow cytometry assays revealed that BADGP- and PUGNAc-exposed GCs exhibited higher JC-1 monomer ratios than that of the control samples (*p* < 0.05; Figure 5A,B). Then, the intracellular ATP level was examined in order to determine the influence of altered O-GlcNAc levels on the energy state of GCs. The results showed that the ATP levels in treated cells were significantly reduced compared with that of the control group (*p* < 0.05; Figure 5C). The intermediate metabolic enzymes involved in the TCA cycle, succinate dehydrogenase (SDH) and malate dehydrogenase (MDH), were also analyzed to gain a better understanding of the role of dysregulated O-GlcNAc states in energy metabolism of bovine GCs. An increased enzymatic activity of SDH was observed in BADGP- or PUGNAc-treated cells, whereas the MDH activity was much lower following treatment with PUGNAc than that of the control and BAGDP-treated groups (*p* < 0.05; Figure 5D,E).

### 2.6. Disruption of O-GlcNAc Cycling Changes the Expression of Thioredoxin-Interacting Protein (TXNIP)

In an attempt to identify the potential mechanism by which the disrupted O-GlcNAc cycling affected the GCs’ glucose metabolism, the expression level of TXNIP, a key regulator of glucose uptake, was measured. As shown in Figure 6, TXNIP mRNA and protein levels were markedly induced in response to BADGP exposure compared to the control group (*p* < 0.05), whereas no significant alteration was observed when PUGNAc was present (*p* > 0.05).

## 3. Discussion

Increasing amounts of data from studies on animals and cell lines suggest that cells are sensitive to the perturbation of O-GlcNAc homeostasis due to altered nutritional availability and metabolic flux, which influence the cellular functions accordingly [26,27,28]. Blocked O-GlcNAc cycling by selective loss of OGA in hematopoietic stem cells impairs the stem cell self-renewal and repopulation capacity [29]. A constitutive OGA knockout mouse exhibits metabolic disorders, perinatal lethality, organ defects, and tissue-specific dysregulation of O-GlcNAc homeostasis [19,30], whereas OGT depletion decreases the proliferation of embryonic neural stem cells and inhibits the migration of newborn neurons [31]. Human cell lines treated with an OGA inhibitor exhibit downregulated OGT protein expression and upregulated OGA protein expression, which indicates that O-GlcNAc homeostasis is correlated with the expression of OGT and OGA [26]. In cancer cells, OGA transcript levels displayed compensatory variations upon alterations in O-GlcNAc status, whereas augmenting overall cellular O-GlcNAc levels leads to decreased transcript and translational efficiency of OGT [28,32]. In the present study, repression of the O-GlcNAc level in GCs via the OGT inhibitor, BADGP, reduced the expressions of the OGT mRNA and protein, as well as the OGA protein. Enhancement of the O-GlcNAc level via the OGA inhibitor, PUGNAc, decreased the expression of the OGT mRNA and protein, but promoted OGA expression at the mRNA and protein level. Our findings are consistent with some previous reports which imply the feedback modulation between variations of O-GlcNAc levels and OGT or OGA expression [26,28]. Intriguingly, pharmacological inhibition of OGT did not lead to a compensatory induction of its own expression. Instead, its expression was significantly inhibited. This observation is inconsistent with those of a previous study conducted on tumor cells [32]. Several studies have claimed that decreased O-GlcNAc levels are not necessarily accompanied by a particular increase in OGT abundance in all cell types [26,32]. As the mechanisms for maintenance of cellular O-GlcNAc homeostasis may vary in tumorous and non-tumorous cells [28], these seemingly discrepant findings observed in the current study further verify the notion that modulation of OGT expression for maintaining balanced O-GlcNAc levels for homeostasis depends on the cell type [32]. However, further studies are required to clarify the exact mechanism involved in this process.

During follicular development, GC proliferation and apoptosis play a critical role in maintaining oocyte growth and ovarian functions [33]. There is accumulating evidence that the disturbance of O-GlcNAc regulation may induce alterations in the cell cycle and cellular proliferation [27]. Previous studies have shown that the deletion of OGT resulted in the loss of embryonic stem cell (ESC) viability [18], caused T-cell apoptosis and fibroblast growth arrest [34], and reduced proliferation and self-renewal of ESCs [35]. In contrast, the disruption of OGA impaired cell proliferation, caused mitotic defects, and led to the deregulation of genes linked to cell proliferation and metabolism [30,36]. Importantly, aberrant O-GlcNAcylation is associated with proliferative diseases such as cancer [27,37]. The above results demonstrated that both OGT and OGA play key roles in the proliferation process of normal and cancer cells. In the current study, we observed that altered O-GlcNAc cycling in response to OGT or OGA pharmacological inhibition reduced cell viability, decreased the expression levels of the proliferation-associated genes *CDC42* and *PCNA*, induced apoptosis, and caused the upregulated *BAX/BCL-2* mRNA ratio and *CASPASE-3* transcripts, which corroborates the findings of a previous study showing that the accumulation of O-GlcNAc in mouse embryonic neural precursor cells induced by treatment with the OGA inhibitor, PUGNAc, activated Caspase-3 activity and severely damaged the cell viability [38]. However, this is slightly different from a recent study conducted on GCs of bovine antral follicles, which revealed that the manipulation of O-GlcNAc by inhibiting the HBP or via the OGT inhibitor, OSMI-1, impairs GC proliferation, but perturbation of O-GlcNAc via the OGA inhibitor, Thiamet-G, has no obvious effect on GC proliferation [24]. This discrepancy may be due to the different OGA inhibitor used. Thiamet-G is an efficient transition-state mimic that engages in extensive interactions with OGA active-site residues, whereas PUGNAc is a catalytic transition-state analogue, and inhibits both OGA and other glycoside hydrolases, including α-glucosaminidases and lysosomal β-hexosaminidases [39]. It has been shown that PUGNAc could inhibit enzymes from other enzyme families and impact cell growth [40]. As the regulatory system of O-GlcNAc cycling is complicated, the off-target effects of inhibitors and the spatiotemporal regulation of cellular processes should be taken into account to yield more precise mechanistic insights.

Glucose uptake and metabolism in GCs supply essential energy substrates and intermediates to ensure oocyte maturation, and disturbance of the metabolic function of GCs leads to abnormalities in follicle development [41,42,43]. Glucose is metabolized through the glycolytic pathway, pentose phosphate pathway (PPP), HBP, and polyol pathway [23]. Among the four identified metabolic pathways, glycolysis accounts for a great proportion of glucose metabolism in GCs [44]. Glycolysis-related proteins were upregulated in GCs during the primordial-to-primary follicle transition, indicating glycolytic activity in GCs is vital for the development of growing follicles. [45] Several studies have presented evidence directly linking O-GlcNAcylation to the regulation of key glycolytic enzymes [10]. For instance, O-GlcNAcylation inhibits phosphofructokinase 1 (PFK1) activity and redirects glucose flux through the PPP, which provides the reducing power critical for cancer cell proliferation and survival [46]. O-GlcNAcylation also represses pyruvate kinase M2 (PKM2) activity and promotes aerobic glycolysis and tumor growth [47,48]. In this study, we observed that the perturbation of O-GlcNAc cycling via BADGP or PUGNAc in GCs significantly decreased HK and PK activities and pyruvate and lactate production, but LDH activity was oddly increased. Abnormal elevation of LDH activity was inexplicable at this moment, but our results indicated that aberrant O-GlcNAc cycling damaged the glycolytic activity of GCs.

Under oxygenated conditions, normal cells utilize glycolysis to metabolize glucose into pyruvate for the production of energy via the TCA cycle and oxidative phosphorylation [10]. Previously, several important intermediate metabolic enzymes involved in the TCA cycle and oxidative phosphorylation were shown to be O-GlcNAcylated [49,50]. Diminishing O-GlcNAcylation levels through OGT knockdown with RNAi resulted in an increase in TCA cycle metabolites [51]. Overexpression of OGT or OGA caused significant decreases in cellular respiration and glycolysis as well as a reduction in several TCA cycle proteins, which implies that mitochondrial function is sensitive to O-GlcNAc cycling [52]. In the current study, we identified a marked decrease in MMP and ATP production in response to changes in O-GlcNAc homeostasis, which is in agreement with those of a recent study conducted on murine embryonic fibroblasts [53]. Furthermore, we observed that pharmacological inhibition of OGA led to a decrease in MDH activity, but the variation of SDH activity was intriguingly increased by alterations in O-GlcNAc cycling. These seemingly paradoxical discoveries imply that there exists a dynamic regulation of the TCA cycle enzymes in response to altered O-GlcNAcylation. However, the exact molecular machinery at work during this process warrants a deeper analysis.

TXNIP has been recognized as a negative regulator of glucose uptake and aerobic glycolysis with pivotal roles in maintaining cell integrity by participating in proliferation, inflammation, and cellular metabolism [54,55]. A previous study showed that under high glucose concentrations, TXNIP can be modified by O-GlcNAc in pancreatic β cells [56]. In leucocytes from patients with type 2 diabetes, the TXNIP mRNA level was significantly correlated with OGA, but not with OGT mRNA [57]. In cancer cells, the glycolytic inhibitor 2-deoxyglucose (2DG) induced TXNIP transcription, and this process is partially due to the inhibition of OGA and an increase in the cellular O-GlcNAc modified proteins [58]. However, 2DG inducing TXNIP expression is dependent on increased O-GlcNAcylation in HEK293T, A549, and C2C12, but not in HEPG2 cells. The possible explanation for this result is the cell line specific effect. Upon 2DG exposure, different cell lines exhibit distinct metabolic characteristics or differences in the HBP flux, which may affect the expression of TXNIP [58]. In the present study, a negative correlation was observed between O-GlcNAcylation and TXNIP expression in bovine GCs upon inhibition of OGT, but repression of OGA had no effect. Our results further demonstrate that the regulation of TXNIP by O-GlcNAcylation is dependent on the cell type.

## 4. Materials and Methods

### 4.1. GC Isolation and Culture

Bovine granulosa cells were isolated and cultivated according to the methods described in our previously published article [59]. Briefly, bovine ovaries were collected from the local abattoir and transported back to the laboratory within 2 h. The collected ovaries were washed 2–3 times repeatedly with sterile saline containing 100 U/mL of penicillin and 0.1 mg/mL of streptomycin, followed by a 2 min wash with 75% ethanol. Then, the ovaries were rewashed with sterile physiological saline containing 100 U/mL of penicillin and 0.1 mg/mL of streptomycin to remove the residual alcohol. To isolate GCs, the follicular fluid was aspirated from follicles with diameters ranging from 2–6 mm using an 18-gauge needle and collected into 15 mL centrifuge tubes (Corning Inc., Corning, NY, USA). The aspirated follicular fluid was filtered through a 40 μm filter to remove the oocyte–cumulus complexes and cellular debris, and the liquid containing GCs was centrifuged twice at 1000 rpm for 5 min. GC pellets were resuspended in 1 × red blood cell lysis buffer (Biovision, Milpitas, CA, USA) to lyse erythrocytes, followed by the addition of DMEM/F-12 medium containing 10% FBS (Thermo Fisher Scientific, Boston, MA, USA) to terminate the lysis buffer reaction. Approximately, a density of 5 × 10^5^ cells/mL were seeded in a 6-well plate, and cultured in DMEM/F-12 medium supplemented with 10% FBS, 100 U/mL of penicillin, and 0.1 mg/mL of streptomycin at 37 °C, 5% CO_2_, and saturated humidity. The medium was replaced every 48 h of incubation until 80–90% confluency was reached.

### 4.2. RNA Extraction and Quantitative Real-Time PCR (qRT-PCR)

Total RNA was extracted from collected cells using an RNAprep pure cell kit (TIANGEN, Beijing, China) according to the manufacturer’s instructions. The quality and concentration of the RNA samples were assessed by measuring the absorbance at 230, 260, and 280 nm with a spectrophotometer (BioDrop μLite, Cambridge, UK). The extracted RNAs were of high purity, with an A_260_/A_280_ ratio between 2.01 and 2.11, and an A_260_/A_230_ ratio between 2.03 and 2.25. The cDNA was synthesized from total RNA using a PrimeScript RT reagent kit (Thermo Fisher Scientific, Boston, MA, USA). qRT-PCR analysis was conducted in triplicate using an ABI-7900 SDS instrument (Applied Biosystems, Foster City, CA, USA) according to the methods described in our previously published paper [60]. The primers used are listed in Table 1. The NCBI Primer-Blast tool was used for primer design based on the reference sequences of each gene, and the quality of the gene primers was validated by agarose gel electrophoresis. The 2−∆∆CT method was used to calculate the gene expression level, using GAPDH as a reference gene.

### 4.3. Protein Extraction and Western Blot Analysis

After treatment, GCs from each group were placed on ice and washed three times with 1 mL of PBS prior to lysis in a RIPA buffer (Beyotime, Shanghai, China). The lysates were vortexed for 15 s and centrifuged at 20,000 rpm for 10 min at 4 °C to remove the cell debris. Total protein concentration was measured using an enhanced BCA protein assay kit (Beyotime, Shanghai, China). After that, samples were denatured by boiling at 100 °C for 10 min, and frozen at −80 °C until used. Western blot analysis was conducted as previously described [61] with minor modifications. Briefly, the protein samples were separated on 4–15% gradient SDS-PAGE gels according to the standard procedures. After that, samples were transferred onto a nitrocellulose membrane (BioTrace NT, Pall Corp., Port Washington, NY, USA). Membranes were blocked with 5% (*w*/*v*) nonfat milk in Tris-buffered saline (TBS) containing 0.1% Tween 20 (TBST) at room temperature for 1 h. The blots were then incubated overnight at 4 °C with primary antibodies against O-GlcNAc (1:1000; NB300-524; Novus, Littleton, CO, USA), OGT (1:1000; ab96718; Abcam, Cambridge, UK), OGA (1:1000; ab105217; Abcam, Cambridge, UK), TXNIP (1:1000; ab188865; Abcam, Cambridge, UK), and β-actin, (1:1000; 4967 s; Cell Signaling Technology, Beverly, MA, USA). After three TBST washes, the membranes were incubated with horseradish peroxidase (HRP)-conjugated secondary antibodies (HRP-labeled goat anti-mouse (1:5000), and goat anti-rabbit (1:5000) (Beyotime, Shanghai, China)) for 1 h at room temperature. Afterwards, the blots were washed and detected using an enhanced chemiluminescent detection kit (Tanon, Shanghai, China) according to the manufacturer’s instructions. The intensities of the bands were measured using ImageJ 1.44p software (National Institutes of Health, Bethesda, MD, USA).

### 4.4. Cell Viability Analysis

The CellTiter-Lumi^TM^ Plus Luminescent cell viability assay kit (Beyotime, Shanghai, China) was used to determine the cell viability following the manufacturer’s protocol. Briefly, GCs were seeded in a 96-well plate at a density of 1 × 10^5^ cells/well, and incubated overnight at 37 °C in humidified air containing 5% CO_2_. The cells were equilibrated for 10 min at room temperature. Then, 100 μL of the CellTiter-Lumi^TM^ Plus detection reagent was added to each well, and vibrated for 2 min at room temperature to lyse the samples. After an additional 10 min incubation at room temperature, the luminescent signals were measured by a fluorescence microplate reader (Tecan, Shanghai, China).

### 4.5. Apoptotic Assay

The apoptotic assay was conducted using the Annexin V-FITC apoptosis detection kit (Beyotime, Shanghai, China) following the manufacturer’s protocols. Briefly, GCs were harvested and washed twice with cold PBS, then resuspended in 195 μL of FITC-conjugated annexin V solution at a final concentration of 1 × 10^5^ cells/well. The cell samples were added to 5 μL of annexin V-FITC, vortexed gently, and added to another 10 μL of PI, followed by mixing. The mixture was incubated in the dark at room temperature for 15 min, and cooled on ice. The apoptotic rate was measured using a flow cytometer (BD Biosciences, San Jose, CA, USA).

### 4.6. Assessment of MMP

The MMP of GCs was detected using the MitoProbe™ JC-1 Kit (Thermo Fisher Scientific, Boston, MA, USA) according to the manufacturer’s instructions. Briefly, after three washes in PBS, GCs were stained with 4 µM of JC-1 dye for 30 min in the dark at 37 °C, 5% CO_2_ in saturated humidity. The samples were sorted out using a flow cytometer (BD Biosciences, San Jose, CA, USA).

### 4.7. Enzymatic Activity Assay

GCs were rinsed twice with 0.1% PVA/PBS and stored at −80 °C until needed. Prior to measurement, all GC samples were centrifuged at 4000 rpm for 10 min, and then the lower precipitated cells were homogenized with 40 μL of double-distilled water and immediately used for the following experiments.

HK activity was measured using a commercial PK assay kit (Nanjing Jiancheng Bioengineering Institute, Nanjing, China) according to the manufacturer’s instructions. Briefly, 30 μL of cell homogenate, 400 μL of Reagent I, 400 μL of Reagent II, 80 μL of Reagent III, 80 μL of Reagent IV, 40 μL of Reagent V, and 8 μL of Reagent VI were added to the centrifuge tube and mixed thoroughly. The optical density (OD) values were measured immediately at a wavelength of 340 nm using a spectrophotometer (Unico 7200, Texas, USA).

PK activity was measured using a commercial PK assay kit (Nanjing Jiancheng Bioengineering Institute, Nanjing, China) according to the manufacturer’s instructions. Briefly, 10 μL of samples, 1 mL of Reagent I, 50 μL of Reagent II, 50 μL of Reagent III, 250 μL of Reagent IV, and 50 μL of Reagent V were added to the centrifuge tube and mixed thoroughly. The OD values were measured immediately at a wavelength of 340 nm.

LDH activity was measured using a commercial assay kit (Nanjing Jiancheng Institute of Biological Engineering, Nanjing, China) according to the manufacturer’s protocol. Briefly, 20 μL of cell homogenate, 25 μL of buffer solution, and 5 μL of coenzyme I were added to the centrifuge tube and mixed thoroughly. Then, after incubation at 37 °C for 10 min, 25 μL of 2,4-dinitrophenylhydrazine was added and the mixture was incubated for another 10 min at 37 °C, followed by adding 250 μL of 0.4 M NaOH solution and incubating for another 5 min at room temperature. The OD values were measured at a wavelength of 450 nm.

SDH activity was detected using a commercial assay kit (Nanjing Jiancheng Institute of Biological Engineering, Nanjing, China) following the manufacturer’s protocol. Specifically, 100 μL of cell homogenate and 2.6 mL of the working solution were added to a 5 mL pipette, and mixed immediately. The OD values were measured at a wavelength of 600 nm using a spectrophotometer.

MDH activity was measured using a commercial assay kit (Nanjing Jiancheng Institute of Biological Engineering, Nanjing, China) according to the manufacturer’s instructions. Briefly, 50 μL of cell homogenate, and 1 mL of the working solution were added to the test tube, and vortexed thoroughly. The OD values were measured at a wavelength of 340 nm using an ultraviolet spectrophotometer.

### 4.8. Metabolites Measurement and Intracellular ATP Level Assays

Pyruvate production by the GCs was measured using a commercially available assay kit (Solarbio, Beijing, China) according to the manufacturer’s instructions. Specifically, follicular GCs were collected into centrifuge tubes at a density of 1 × 10^4^ cells/mL, and mixed with 500 μL of extract I. After ultrasonication and centrifugation (8000 rpm for 10 min), the supernatant was recovered and stored on ice until analysis. Then, 75 μL of standard solution or cell homogenate was added to a well in a 96-well plate, and mixed with 25 μL of Reagent I. The mixture was incubated at room temperature for 2 min, then 125 μL of Reagent II was added and mixed. The OD value was measured at a wavelength of 520 nm.

Lactate content was measured using a commercially available assay kit (Solarbio, Beijing, China) according to the manufacturer’s instructions. Briefly, 10 μL of standard solution or cell homogenate, 40 μL of Reagent I, 10 μL of Reagent II, and 20 μL of Reagent IV were added to a centrifuge tube and mixed thoroughly. After incubation in a water bath at 37 °C for 20 min, 60 μL of Reagent III and 6 μL of Reagent V were added to each tube, and incubated at 37 °C for 20 min in the dark. The OD value was measured at a wavelength of 570 nm.

A commercial ATP assay kit (Solarbio, Beijing, China) was used to detect changes in the ATP level of the GCs following the manufacturer’s instructions. In detail, follicular GCs were collected and lysed in 500 μL of extract Reagent I. Afterward, 500 μL of supernatant was mixed with 500 μL of chloroform, and placed on ice for measurement. Then, 20 μL of standard solution or cell homogenate, 128 μL of Reagent I, and 52 μL of working solution were added to a centrifuge tube and mixed thoroughly. The OD value was measured at a wavelength of 340 nm.

### 4.9. Statistical Analysis

All the experiments were repeated, and average data were expressed as mean ± SEM of at least three biological replicates. Statistical analysis was carried out by a one-way ANOVA, followed by multiple comparison post hoc tests using the statistical analysis system software 9.0 (SAS Institute, Cary, NC, USA) and GraphPad Prism 7 statistical software (GraphPad Software Inc., San Diego, CA, USA). Differences were considered to be statistically significant when the *p* value was less than 0.05.

## 5. Conclusions

Collectively, our results verify the notion that a fine-tuning intracellular O-GlcNAc homeostasis is maintained by the feedback regulation of OGT and OGA expression. The perturbation of O-GlcNAc cycling has a profound effect on GC function and glucose metabolism (Figure 7). Our findings provide some meaningful information that can be applied for female infertility in metabolic diseases caused by abnormal O-GlcNAcylation. Moreover, with the development of new tools that regulate OGT or OGA activities with spatiotemporal precision in cells, additional details will be elucidated in future studies in order to better understand the O-GlcNAc function in bovine granulosa cells.

## Figures and Tables

**Figure 1 ijms-23-07815-f001:**
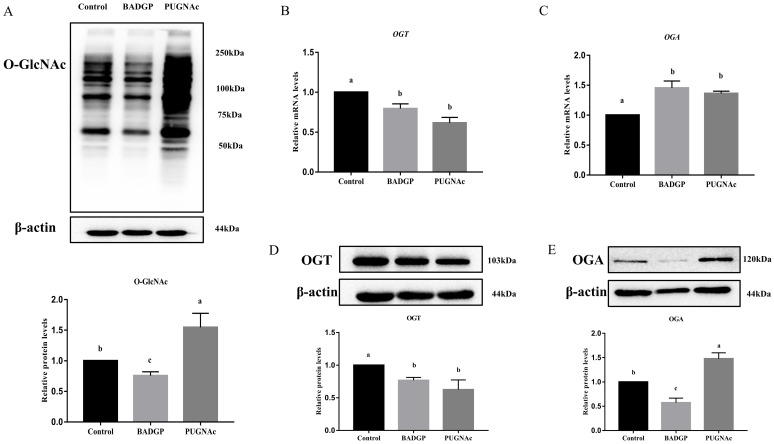
Alteration of O-GlcNAc levels affects the expression of OGT and OGA in bovine granulosa cells. (**A**) Immunoblot of overall O-GlcNAc levels, and the chart below shows the relative amounts of proteins. (**B**) *OGT* mRNA levels. (**C**) *OGA* mRNA levels. (**D**) Immunoblot of OGT levels, and the chart below shows the relative amounts of proteins. (**E**) Immunoblot of OGA levels, and the chart below shows the relative amounts of proteins. Values are mean ± SEM of at least three replicates. Different superscript letters (a–c) indicate significant differences (*p* < 0.05).

**Figure 2 ijms-23-07815-f002:**
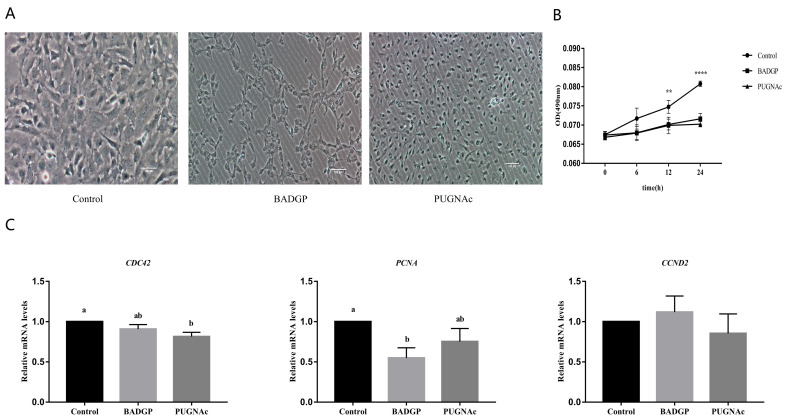
Disruption of O-GlcNAc cycling affects viability and proliferation of GCs. (**A**) Representative photomicrographs of GCs. Bar = 100 μm. (**B**) Effects of dysregulated O-GlcNAcylation on cell viability at different time points. Cell viability was tested in samples challenged with BADGP or PUGNAc at 6 h, 12 h, and 24 h, respectively. (**C**) Relative mRNA expression levels of proliferation-related genes *CDC42*, *PCNA*, and *CCND2* in GCs. Values are mean ± SEM of at least three replicates. Different superscript letters (a, b) indicate significant differences (*p* < 0.05). ** *p* < 0.01, **** *p* < 0.0001.

**Figure 3 ijms-23-07815-f003:**
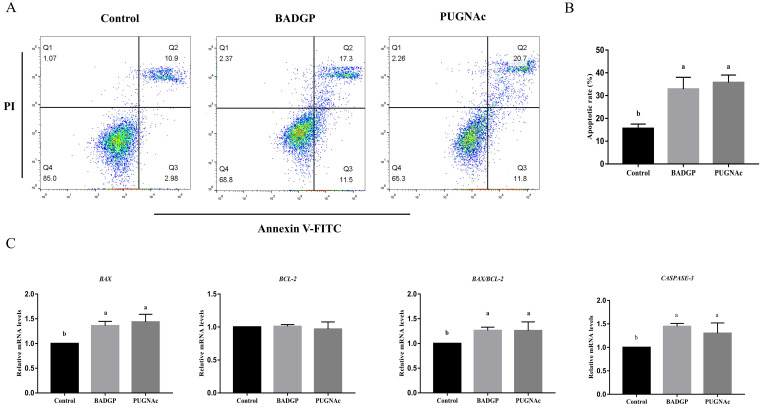
Disruption of O-GlcNAc cycling induces cell apoptosis in bovine GCs. (**A**) Annexin V/PI flow cytometric analysis of GCs upon pharmacological inhibition of OGT and OGA. GCs were labeled with annexin V-FITC and PI. Necrotic cells, late apoptotic cells, early apoptotic cells, and viable cells are shown in the Q1, Q2, Q3, and Q4 area, respectively. (**B**) Effects of dysregulated O-GlcNAcylation on cell apoptosis in GCs. (**C**) Relative mRNA expression levels of apoptotic-related genes *BAX*, *BCL-2*, and *CASPASE-3* in GCs. Values are mean ± SEM of at least three replicates. Different superscript letters (a, b) indicate significant differences (*p* < 0.05).

**Figure 4 ijms-23-07815-f004:**
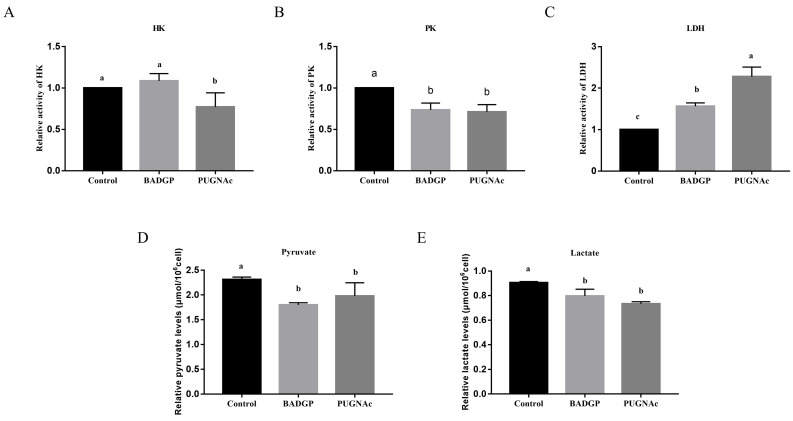
Perturbation of O-GlcNAc cycling in bovine GCs reduces glycolysis. (**A**) Hexokinase (HK) activity. (**B**) Pyruvate kinase (PK) activity. (**C**) Lactate dehydrogenase (LDH) activity. (**D**) Pyruvate production. (**E**) Lactate production. Values are expressed as mean ± SEM of at least three replicates. Different superscript letters (a–c) indicate significant differences (*p* < 0.05).

**Figure 5 ijms-23-07815-f005:**
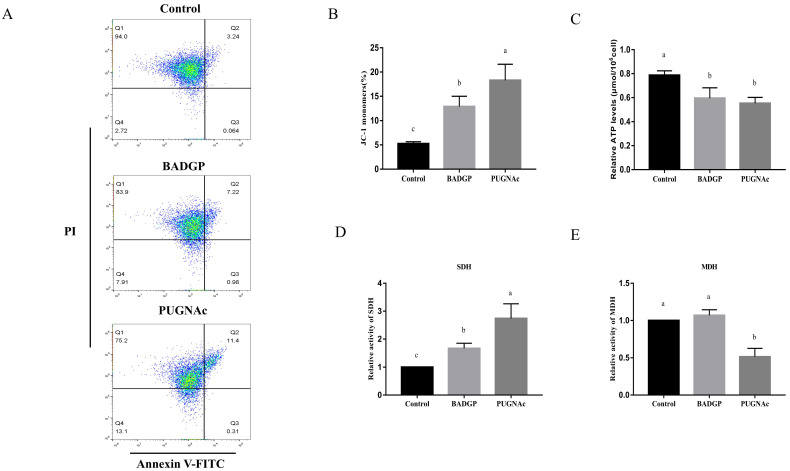
Alteration in O-GlcNAc levels impairs mitochondria homeostasis, ATP production, and the tricarboxylic acid (TCA) cycle. (**A**) Flow cytometric analysis of mitochondrial membrane potential (MMP) in GCs. GCs were stained with JC-1, a reporter molecule for heterogeneity in MMP in living cells. (**B**) Effects of dysregulated O-GlcNAcylation on MMP in GCs. (**C**) Intracellular ATP levels. (**D**) Succinate dehydrogenase (SDH) activity. (**E**) Malate dehydrogenase (MDH) activity. Values are expressed as mean ± SEM of at least three replicates. Different superscript letters (a–c) indicate statistically significant differences (*p* < 0.05).

**Figure 6 ijms-23-07815-f006:**
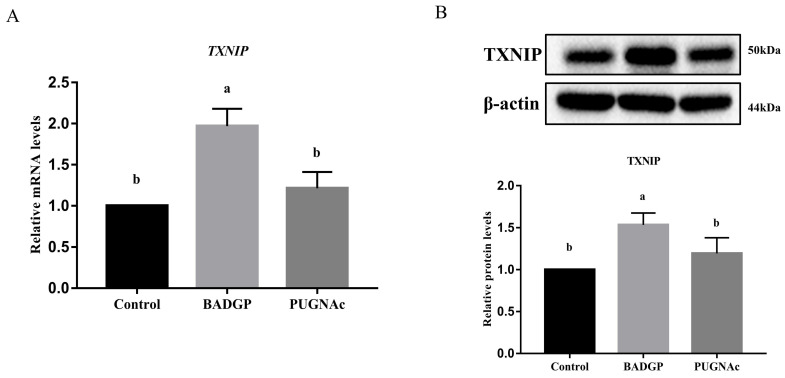
Disruption of O-GlcNAc cycling changes the expression of TXNIP. (**A**) *TXNIP* mRNA levels by qRT-PCR analysis. (**B**) TXNIP protein levels by Western blot analysis. Values are expressed as mean ± SEM of at least three replicates. Different superscript letters (a, b) indicate statistically significant differences (*p* < 0.05).

**Figure 7 ijms-23-07815-f007:**
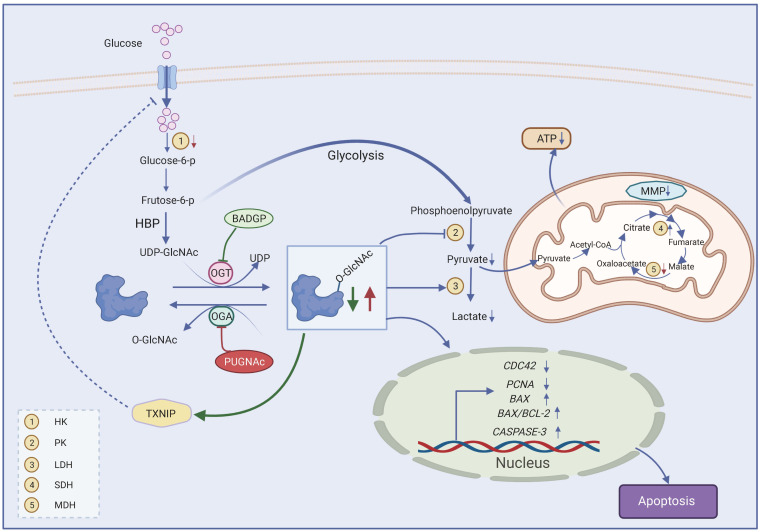
Proposed model for effects of dysregulated O-GlcNAcylation on GC function. Frutose-6-p, fructose-6-phosphate; HBP, hexosamine biosynthetic pathway; O-GlcNAc, O-linked β-N-acetylglucosamine; OGT, O-GlcNAc transferase; OGA, O-GlcNAcase; UDP-GlcNAc, uridine-5′-diphosphate N-acetylglucosamine; BADGP, benzyl-2-acetamido-2-deoxy-α-D-galactopyranoside; PUGNAc, O-(2-acetamido-2-deoxy-D-gluco-pyranosylidene); TXNIP, thioredoxin-interacting protein; TCA cycle, tricarboxylic acid cycle; MMP, mitochondrial membrane potential. Created with BioRender.com, accessed on 11 May 2022.

**Table 1 ijms-23-07815-t001:** Primer sequences used for qRT-PCR.

Gene	Primer	Sequence	Accession No.
*GAPDH*	Forward	5′-GGGTCATCATCTCTGCACCT-3′	NM_001034034
	Reverse	5′-GGTCATAAGTCCCTCCACGA-3′	
*OGT*	Forward	5′-AGGGTTCGAAGGCTGTAACTG-3′	NM_001098070
	Reverse	5′-ACTCAGCTAACCCTGTGCTG-3′	
*OGA*	Forward	5′-TTGAAGAATGGCGGTCACGA-3′	NM_001206448
	Reverse	5′-TGACTACGACACCCTAACCAC-3′	
*CDC42*	Forward	5′-GTTGTTGTGGGTGATGGTGC-3′	NM_001046332
	Reverse	5′-TCCCCACCAATCATAACTGT-3′	
*PCNA*	Forward	5′-GTCCAGGGCTCCATCTTGAA-3′	NM_001034494
	Reverse	5′-CAAGGAGACATGAGACGAGT-3′	
*CCND2*	Forward	5′-TGACCGCTGAGAAGTTATGC-3′	NM_001076372
	Reverse	5′-CGCCAGGTTCCATTTCAACT-3′	
*BAX*	Forward	5′-GGCTGGACATTGGACTTCCTTC-3′	NM_173894
	Reverse	5′-TGGTCACTGTCTGCCATGTGG-3′	
*BCL-2*	Forward	5′-GAGTCGGATCGCAACTTGGA-3′	NM_001077486
	Reverse	5′-CTCTCGGCTGCTGCATTGT-3′	
*CASPASE-3*	Forward	5′-TACTTGGGAAGGTGTGAGAAAACTAA-3′	NM_001077840
	Reverse	5′-AACCCGTCTCCCTTTATATTGCT-3′	
*TXNIP*	Forward	5′-TGGACTACTGGGTGAAGGCT-3′	NM_001101875
	Reverse	5′-AGCTGACACAGGTTCCAGTAAAT-3′	

## Data Availability

The data used to support the findings of this study are available from the corresponding author upon reasonable request.

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
