# Peer review of "Disruption of O-GlcNAcylation Homeostasis Induced Ovarian Granulosa Cell Injury in Bovine"

_ijms, 2022, doi:10.3390/ijms23147815_

Round 1
Reviewer 1 Report
This manuscript details the role of O-GlcNAcylation levels in bovine ovarian granulosa cells, in terms of cell proliferation and apoptosis as well as metabolic intermediates and enzyme activities. The manuscript is well written and covered a broad spectrum of processes downstream of O-GlcNAcylation; however, the manuscript could be strengthened through additional information in the methods section and a full analysis in the discussion section of potential off target effects of the two inhibitors used in the study.
Major Comments:
Methods:
- Neither BADGP or PUGNAc are specific inhibitors of OGT or OGA, respectively. BADGP inhibits glycosyltransferases in general (including N-glycosylation) and reduces the flux of the hexosamine biosynthetic pathway. PUGNAc inhibits a variety of N-acetylhexosaminidases and impacts insulin signaling. Due to the lack of specificity, the results pertaining to cell proliferation/apoptosis as well as metabolic intermediates/enzyme activities cannot be concluded to be specific to O-GlcNAcylation levels. This contingency should be noted in discussion and potential roles of off-target effects should be considered in interpretation of the results. Additionally, justification for the use of these specific inhibitors would be beneficial.
- Several portions of the methods were incomplete, including the following:
o Was granulosa cell purity assessed?
o Why was a 24 hr culture chosen?
o How were qPCR primers designed and validated?
Minor Comments:
Ln 13: change “cells” to “cell”
Ln 16: change “decreases of” to “decreased”
Ln 68: specify that the species is bovine
Ln 77-78: are both the BADGP and PUGNAc from Sigma?
Ln 81: it would be helpful to include the preliminary study data as supplemental information
Ln 210-216: Expand on this to provide context as to why the results from this study are inconsistent with those seen in tumor cells
Ln 221: change “respond” to “response”
Ln 230-234: expand on this to provide context as to why the results from this study are inconsistent with proliferation results from previous studies – if the authors hypothesize off target effects are responsible, note potential off target effects that would lead to the discrepancy in proliferation rates
Ln 250-251: what could potentially lead to elevated LDH activity
Ln 265: what could potentially lead to elevated SDH activity
Ln 278: change “depend” to “dependent”
Ln 275-278: expand on this provide context as to why the results from this study are inconsistent with trends from previous studies
Ln 298-299: was on-column DNase treatment performed
Ln 300: was a kit used to assess quality and concentration of RNA?
Ln 301: was a reaction control without reverse transcriptase (No RT) performed?
Reviewer 2 Report
I noticed that beta-actin bands are repeated in different figures and the authors did not provide the original blots without cropping.
1- Beta-actin of Fig.1B is the same as Fig.1A.
2- Beta-actin in Fig.6B is the same as Fig.1C
Authors are advised to provide the original images with the correct beta-actin bands.
Reviewer 3 Report
In this study, the authors focused on the effect of disrupted O-GlcNAc cycling on the proliferation and apoptosis of bovine granulosa cells and further aimed to determine how this influenced glucose metabolism. They found that perturbation of O-GlcNAc cycling has a deep consequence on granulosa cell function and glucose metabolism.
The manuscript is well written. Data and discussion of the results are convincing. In my opinion, the manuscript fits the standard of the journal.
Round 2
Reviewer 2 Report
The authors responded to my requests.